# Modeling and Optimization of the Para-Xylene Continuous Suspension Crystallization Separation Process via a Morphology Technique and a Multi-Dimensional Population Balance Equation

**Zhenxing Cai, Jixiang Liu** **, Hui Zhao, Xiaobo Chen and Chaohe Yang** *

State Key Laboratory of Heavy Oil Processing, China University of Petroleum, Qingdao 266580, China
* Correspondence: yangch@upc.edu.cn

**Abstract:** In this study, we carried out a para-xylene crystallization experiment at constant temperature and concentration levels. Throughout the process, the kinetics of nucleation, growth, breakage, and aggregation of para-xylene particles were measured and built using a morphological approach. An additional a three-stage continuous suspension crystallization separation experiment was carried out, the process for which was simulated using the population balance model based on correlated kinetic equations. The population balance equation was solved using an extended moment of classes algorithm, and the solving process was implemented in MATLAB. In this case, the predicted particle size distribution of the products matched well with the experiment. In order to provide references for the optimization of the industrial para-xylene crystallization process, a three-stage suspension crystallization separation experiment was designed and conducted, in which each crystallizer had a distinct operating temperature and mean residence time. The effects of operating parameters on the final product were investigated further. The proposed models and algorithms can also be applied in other cases and provide an alternative approach for optimizing continuous crystallization processes.

**Keywords:** crystallization kinetics; morphology; population balance; para-xylene; process optimization



## 1. Introduction

Para-xylene (PX) is the essential raw material in the production of polybutylene terephthalate (PBT) and polyethylene terephthalate (PET) [1], which are the intermediates of polyester fibers, resins, bottles, and films [2]. As such, PX is one of the most important petrochemical materials used in the plastics industry worldwide. PX is mainly derived from the catalytic reforming of crude oil and can co-exist with other C8 isomers including meta-xylene (MX), ortho-xylene (OX), and ethylbenzene (EB) as a mixture. The process of PX production mainly involves two operating steps [1]: isomerization, where other C8 compounds are converted into PX, and separation, where the newly converted pure PX is extracted from the mixture. The separation of PX from the C8 mixture is a difficult process because of the close boiling points and similar molecular structures of the compounds. Distillation, adsorption, and crystallization are the three main approaches for PX separation.

The thermodynamic properties of C8 isomers are given in Table 1, below.

From the early to mid-1900s, the crystallization PX separation method was rarely used due to low PX yield and eutectic point restriction, making the distillation approach the method of choice [3]. The development of adsorption technology ushered in a wave of the method's widespread adoption; distillation remained the most conventional separation method for many years. However, because of the close boiling points present in the method, 150 theoretical plates were needed to separate OX to commercial specifications, while 360 theoretical plates were needed to isolate PX and MX [4]. In the 1960s, Universal Oil Products (UOP) developed a selective adsorption process using Simulated Moving

Bed (SMB) technology [5], which is a continuous chromatographic countercurrent process. SMB separation was accomplished by using the affinity differences of the adsorbent for PX relative to other C8 isomers. The adsorbed PX was then removed from the adsorbent by displacement with a desorbent. Through this process, the purity of the PX product was able to reach 99.7%.

**Table 1.** The thermodynamic properties of C8 isomers from NIST.

| | PX | OX | MX | EB |
|---|---|---|---|---|
| Molecular structure | | | | |
| $T_b$/K | 411.4 | 417.0 | 412.3 | 409.3 |
| $T_m$/K | 286.3 | 248.0 | 225.0 | 179.0 |
| $\Delta H_b$/(kJ/mol) | 42.0 | 42.0 | 41.0 | 41.0 |
| $\Delta H_m$/(kJ/mol) | 17.1 | 13.6 | 11.6 | 9.2 |

where $T_b$: normal boiling point, $T_m$: melting point, $\Delta H_b$: enthalpy of vaporization, $\Delta H_m$: enthalpy of melting.

The PX yield through isomerization remained low until the 1990s, after which the development of new techniques, including UOP's Isomar, Axens' Oparis, and ExxonMobil's XyMax [6], in particular, enhanced the recovery rates. During this time, the crystallization process gained popularity worldwide as feedstock, which can possess PX levels of more than 80% and for which crystallization is more suitable, became widely used. The most common two PX crystallization methods are the following [7]: layer-based melt crystallization and suspension melt crystallization, which are batch and continuous operations, respectively. The equipment structure and operating procedures of layer-based crystallization separation are given in Figure 1. Here, the diagram illustrates how the mixture solution flows down through the inside surfaces of the tubes, allowing the cooling and heating liquids to be distributed evenly across the external surfaces of the tubes. During the crystallization step, a coolant is used to chill the tubes, causing PX to crystallize on the inside surface. Following this, partial melting (sweating) is induced by raising the temperature of the coolant, thus enhancing the purity of the final product. The final melting step is achieved by increasing the temperature. As such, optimum results are achieved through the accurate control of the heating and cooling profiles.

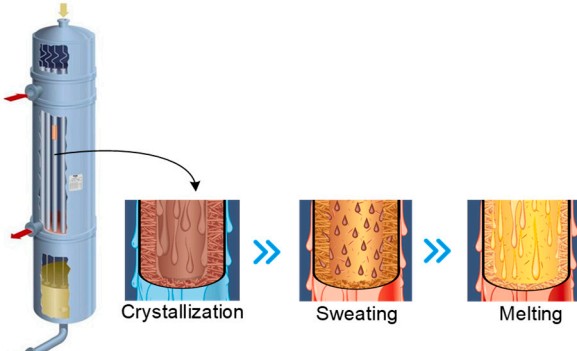

Crystallization    Sweating    Melting

**Figure 1.** The equipment structure and operating procedures of layer-based crystallization.

As layer-based crystallization is a batch operating process, it is less efficient in comparison to the suspension crystallization separation method. Figure 2 illustrates the simplified three-stage PX crystallization separation process. The suspension melt crystallization process is somewhat similar to the cooling crystallization in solution; that is, the mother liquor containing PX is introduced into multi-stage crystallizers with a surface-scraped rotator, and the operating temperatures of multi-stage crystallizers gradually decrease, causing the PX particles to grow in suspension status within the mother liquor [7]. After the last crystallizer, the crystal slurry is centrifuged and filtered to obtain the crude PX crystals.

The remaining mother liquor is then recycled back to the xylene isomerization reactor to enhance the yield, after which the crude PX crystals are re-melted and washed to obtain a high-purity product. Using this method, the purity can reach more than 99.9%.

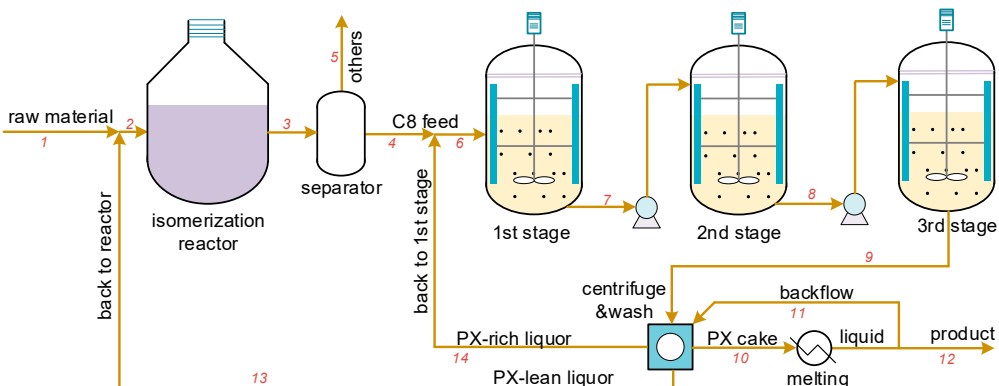

**Figure 2.** The three-stage PX suspension crystallization separation process, where 1-14 are the streams and were used for process optimization in Section 4.2.

The suspension crystallization separation method is a continuous operation that has the advantages of enhanced efficiency, flexibility, and quality [8]. Because of this, PX continuous suspension crystallization separation technology has been widely adopted by commercial companies including Lummus, GTC and Sulzer. The crystallization process includes nucleation, growth, aggregation, and breakage steps, and their respective kinetics are significant to process the simulation and optimization of the crystallization. Patience [9] estimated PX kinetic models' parameters from online measurements of bulk temperature and slurry transmittance in a pilot-scale scraped-surface crystallizer, which resulted in a somewhat unreliable data set. Goede [10] measured the average growth rate of particle size and established the relationship between growth rate and temperature. However, the PX particle is a rectangular plate and has anisotropy in each of its crystal faces, and supersaturation is more suitable to build the equation of growth kinetics. Elsewhere, Mohameed [11] investigated the weight change of PX particles by measuring the solution's concentration in real time. In summary, while the growth kinetics of each face of a PX particle have not been thoroughly reported in published papers, nucleation, aggregation, and breakage kinetics also require further study.

The morphology approach has been widely applied on crystallization process investigation. Huo [12] used an in situ crystal morphology approach to identify the shape and size distribution of L-glutamic acid. Anda [13] adopted an imaging technique for real-time production morphology monitoring. Kodjoe [14] observed the crystallization behavior of an HDPE/CNT nanocomposite via a morphology approach. The growth rate of triclinic N-docosane crystallizing from a N-dodecane solution was measured by the morphology approach in Camacho's work [15]. The morphology approach performs better in characterizing the 3D shape of crystals than traditional methods.

In order to obtain the kinetics of the PX suspension crystallization separation process, as well as to enhance the product yield and reduce the energy cost, a new morphological approach was adopted to investigate the nucleation, growth, aggregation, and breakage behaviors. A high-resolution microscope camera complemented by customized equipment was used to record the growth rate of each facet of PX particles and their change in quantity. Then, the crystallization kinetics were built as a function of supersaturation or crystal size, and the two-dimensional population balance equation was used to simulate the particle size distribution within the crystallizer. A simplified PX isomerization and crystallization separation process is shown in Figure 2, wherein the optimal operating parameters for highest product yield and lowest energy cost can be found by simulating the whole process based on build models.

## 2. Materials and Methods

### 2.1. Materials and Equipment

The compounds chosen for this work were PX, MX, OX, and EB, which were supplied by Aladdin Co., (Shanghai, China) with purity levels of 99.0%, 99.0%, 99.0% and 99.5%, respectively. The purity was confirmed by gas chromatography using a GC-7820 (Huifen Co., Beijing, China). The chemical structures are shown in Table 1. A microscope camera with 4K resolution was purchased from Zhongweikechuang (Shenzhen, China).

### 2.2. Batch Crystallization Experiment

To determine the kinetics of nucleation, growth, breakage, and aggregation at various given concentrations, an experimental apparatus that uses constant temperature and concentration levels was built. The experimental setup and schematic representation are given in Figure 3, and the operating conditions are given in Table 2. The crystallization unit was installed within a low-temperature thermostatic bath, and a magnetic stirrer at a stirring rate of 60 r/min was used to keep the mother liquor well mixed. A gear pump was used to recycle the solution, and the crystallization temperature and solution concentration were kept constant. As the PX particles nucleated and grew within the mother liquor, a microscope camera was used to record the particle growth process. A scale plate was laid on the bottom of the crystallization unit to ensure that the size of particles could be measured. The density of PX particles is higher than that of the liquid, meaning that the particles will settle to the bottom of the solution if the stirring rate is low (as seen in Figure 4, Step 1); thus, a particle can be observed continuously. A cotton ball was installed in the outlet to prevent any accidental loss of particles. The images captured by the camera were transferred to a computer, where the boundaries of the particles and the growth rate in the directions of both width and length were determined, as seen in Figure 4. The change in the quantity of particles was also recorded in real time, in order to correlate the kinetics of nucleation, breakage, and aggregation.

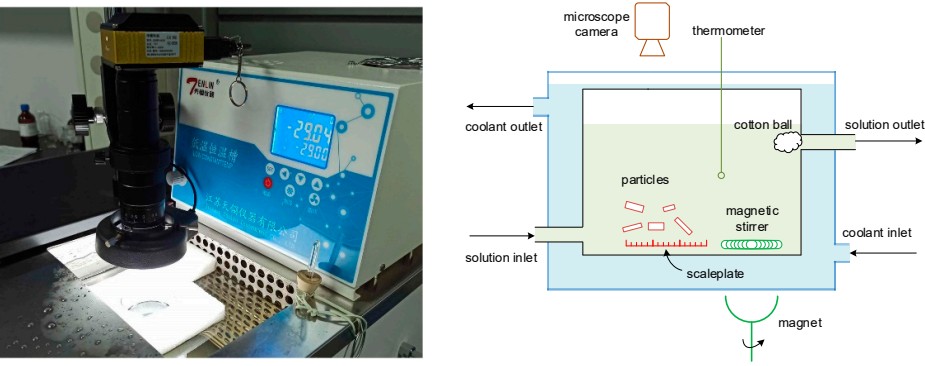

**Figure 3.** The experimental setup and schematic representation for determining the kinetics of nucleation, growth, breakage, and aggregation.

The scaleplate at the bottom was used for camera calibration and particle size determination. It was a grid-lined piece of paper measuring 1 cm × 1 cm, with lines at intervals of 0.1 mm. A custom program, developed using C++ and OpenCV, was used to detect the crosspoints of the grid (see Figure 4, Step 1). First, the captured image was loaded into the program and a section of it was selected for detection. Then, using the HoughLinesP method from OpenCV to fit the grid lines, the threshold value was adjusted manually, which allowed the crosspoints to be calculated using the fitted lines. Not every crosspoint on the grid should be detected, since points in a uniform arrangement with fixed intervals are sufficient. In this case, the distance between selected crosspoints was 1 mm and the number of points was at least 9 (the more the better). These points were called anchor points and any two points in real world coordinates were separated by a fixed distance, but due to the camera distortion, this distance in the captured images was not uniform. The

projection of a three-dimensional world point $(X_w, Y_w, Z_w)$ on a two-dimensional image point $(X_m, Y_m)$ can be described by the pinhole model, for which the details are provided in the literature [16,17].

**Table 2.** The experimental temperatures and concentrations.

| Concentration | | | *T*/K | Concentration | | | *T*/K |
|---|---|---|---|---|---|---|---|
| **PX** | **OX** | **MX** | | **PX** | **OX** | **MX** | |
| 0.500 | 0.156 | 0.345 | 254.35<br>254.15<br>252.15 | 0.851 | 0.046 | 0.103 | 277.15<br>275.15<br>274.22 |
| 0.600 | 0.124 | 0.276 | 262.15<br>261.15<br>260.15 | | | | 273.15<br>271.45<br>271.15 |
| | | | 258.15 | | | | 283.20 |
| 0.701 | 0.093 | 0.206 | 270.15<br>269.15<br>268.15<br>267.15<br>266.25<br>265.35 | 0.950 | 0.016 | 0.035 | 282.24<br>281.15<br>280.15<br>279.15<br>278.15<br>277.15 |
| | | | 264.25 | | | | |

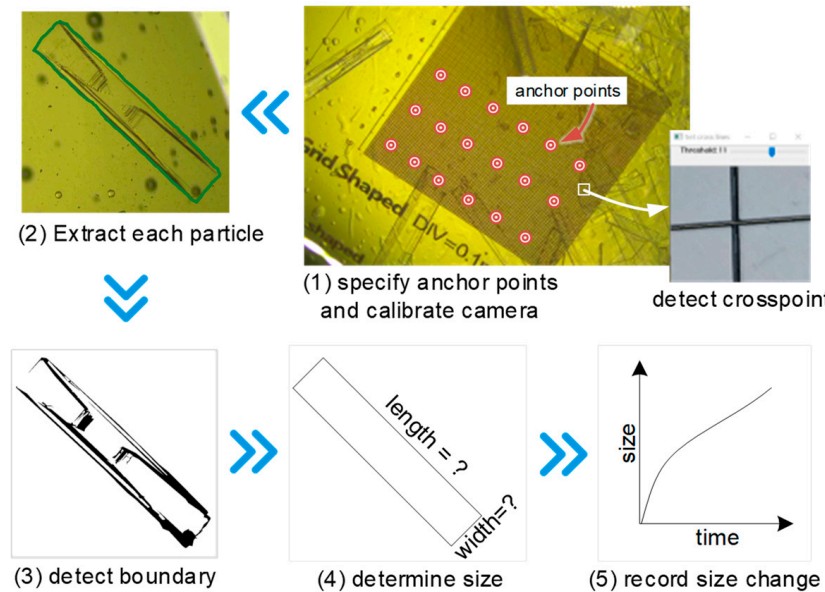

**Figure 4.** The procedure of crystal growth rate measurement.

The projection procedure includes two main steps: the transformation from 3D world to 2D camera coordinates (as represented by the function $f_t$ in Equation (1)), and the simplification of the camera distortion effect on the final 2D image coordinates to the function $f_c$. The scaleplate was assumed to occupy the plane with $Z_w = 0$, wherein the top left anchor point was taken to be the original point, allowing the world coordinates of all the anchor points to be determined. The 2D coordinates of corresponding points on the image were detected over the course of previous steps. With the help of calibrateCamera from OpenCV, the parameters of $f_t$ and $f_c$ were determined. Subsequently, as long as the focal length of the camera was kept constant, any image captured by the camera could be calibrated by initUndistortRectifyMap and remap methods of OpenCV. Essentially, the

camera was calibrated and undistorted images could be retrieved, which could be used for size determination in succeeding sections.

$$\begin{bmatrix} X_m \\ Y_m \end{bmatrix} = f_c \left\{ f_t \left( \begin{bmatrix} X_w \\ Y_w \\ Z_w \end{bmatrix} \right) \right\} \tag{1}$$

After image calibration, cvtColor and threshold methods from OpenCV were applied to smooth and sharpen the image. Due to the crystal's border having a deeper color than the background, the threshold value was adjusted manually so the border could be detected (see Figure 4, Steps 2 and 3). minAreaRect from OpenCV was used to detect the minimum enclosing rectangle (because of PX particle's natural shape), which should overlap with the crystal's border, and when it does not, this implies that the particle has been broken up or aggregated and should not be considered. Then, the length and width of the particle could be determined (as per Figure 4, Step 4) and by detecting a particle continuously, the rate of change in size (i.e., growth rate) could also be derived (as per Figure 4, Step 5). The number of detected crystals was also monitored, and since aggregation and breakage did not occur during the early stages of observation, the change in quantity of crystals could be taken as the rate of nucleation.

PX particles formed as plates and lay on the bottom of the crystallizer when the stirring speed was low, so the above introduced approaches were suitable for determining the width and length of particles in real time, but not for thickness. Therefore, an L-shaped aid was constructed to help measure this dimension (see Figure 5). By adjusting the angle of the fold, the particle could be kept vertical with respect to the camera, allowing measurement. Based on the data points for the population of particles, the mean ratio between thickness and width was found to be 0.113 ($c_t$).

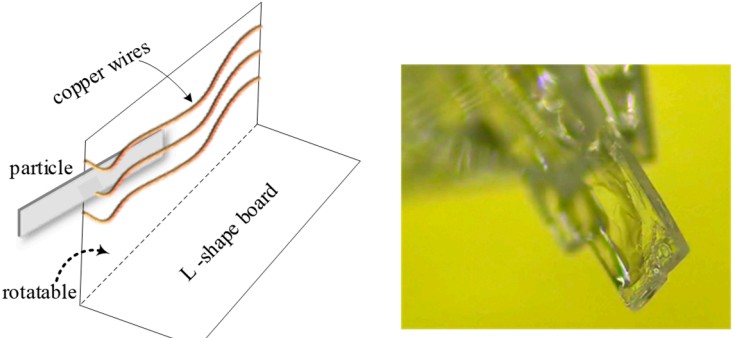

**Figure 5.** The equipment to measure thickness of the particles.

### 2.3. Continuous MSMPR Crystallization

Three MSMPR crystallization units were connected to simulate the industrial PX crystallization process, as shown in Figure 6. The three units each had different operating temperatures and crystallizer volumes (i.e., different mean residence times). The feed was introduced into the first crystallization unit at a given flow rate, the particles nucleated, grew, broke, and aggregated, and then the solution was mixed well using a magnetic stirrer. The solution with the particles was then conveyed to the next unit at the same flow rate, where the nucleation, growth, breakage, and aggregation behaviors continued. The final product was obtained after the third crystallization unit. The particle size distribution in each unit was analyzed using the morphological technique described in Section 2.2. The operating conditions were given in Table 3.

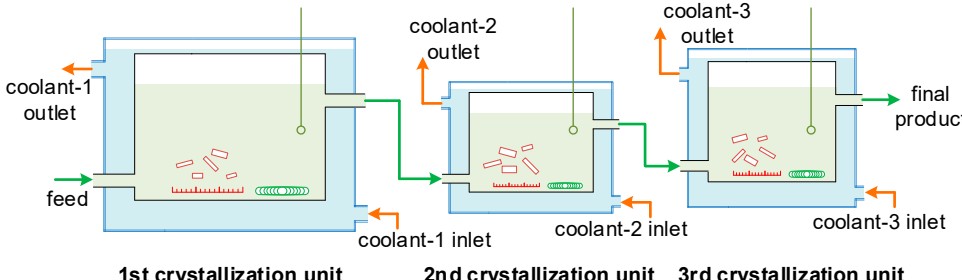

**Figure 6.** The experimental setup for the three-stage MSMPR crystallization process.

**Table 3.** The experimental temperatures and residence time in each crystallizer. $C_0$: the PX concentration in feed, and OX: MX = 0.3; $T$: the operating temperature, K; $t_r$: the mean residence time, s.

| $C_0$ | Crystallizer-1 | | Crystallizer-2 | | Crystallizer-3 | |
|---|---|---|---|---|---|---|
| | $T$/K | $t_r$/s | $T$/K | $t_r$/s | $T$/K | $t_r$/s |
| 0.95 | 282.2 | 20 | 279.2 | 20 | 277.0 | 50 |
| 0.85 | 274.3 | 20 | 271.2 | 20 | 268.2 | 50 |
| 0.70 | 267.8 | 20 | 264.7 | 20 | 262.7 | 50 |
| 0.50 | 256.2 | 20 | 254.2 | 20 | 252.2 | 50 |

## 3. Experimental Results and Models

### 3.1. Nucleation Kinetics

According to the classical nucleation theory, the clusters of particles aggregate together based on the following mechanism [18]:

$$a + a \leftrightarrow a_2$$
$$a_2 + a \leftrightarrow a_3$$
$$a_{i-1} + a \leftrightarrow a_i$$

When the cluster is smaller than the critical size, the force between superficial particles and the surrounding solution is comparable over the intermolecular forces within the cluster. Thus, the cluster has an unstable status. As soon as the cluster grows to a certain size, the intermolecular force prevails and the cluster becomes stable, causing a crystal nucleus to form. Based on the mechanisms of nucleation that occur during this process, nucleation is classified as primary or secondary case. The nucleation rate can be formulated using the fundamental Arrhenius expression, and an expression derived from the Miers nucleation model matches well with experimental data [19]. We adopted the expression proposed by Randolph and Larson [20] in our work, in which the nucleation rate was expressed to be the number of newborn particles per second, as is given in Equation (2).

$$B^0 = k_{\text{nuc}}^1 \left(\frac{c - c^*}{c^*}\right)^{k_{\text{nuc}}^2} \tag{2}$$

where $B^0$ is the nucleation rate, $c$ is the solution concentration, $c^*$ is the saturated concentration at current temperature, and $k_{\text{nuc}}^1$ and $k_{\text{nuc}}^2$ are the two constants. The nucleation experiments were carried out under different temperatures and concentrations, and the process of particle formation was recorded using the microscope camera. The data from the camera were then analyzed and the number change of the particles was detected by a computer. The saturation concentration has been reported in our previous work [21], the binary parameters of activity coefficient model UNIQUAC were fitted and the saturation concentration of mixture with three or more compounds can be predicted. By transforming

Equation (2) into Equation (3), the experimental points were plotted as seen in Figure 7 and correlated by a linear equation. The coefficients were found to be $k_{nuc}^1 = 1.466 \times 10^9$ and $k_{nuc}^2 = 2.280$ with a standard deviation of 0.073.

$$\ln(B^0) = \ln\left(k_{nuc}^1\right) + k_{nuc}^2 \cdot \ln\left(\frac{c - c^*}{c^*}\right) \tag{3}$$

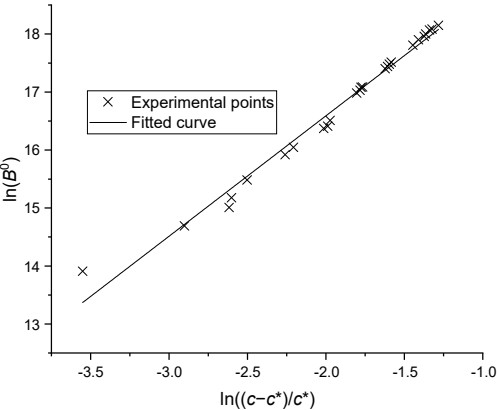

**Figure 7.** The nucleation rates versus supersaturations.

### 3.2. Growth Kinetics

The driving force for crystal growth is the difference in the chemical potential between the current state and equilibrium state, while the surface integration and volume transport phenomena must also be considered [22]. The crystal growth procedure is generally divided into two steps: (1) the molecules break the diffusion boundary layer and diffuse from the bulk solution to the crystal surface; and (2) the molecules distribute on and integrate with the surface of the crystal lattice [23]. However, surface diffusion and desolvation occur before being integrated into the lattice in the second step. The growth rate can be thought as a function of bulk supersaturation, which is a more realistic and applicable mathematical formulation. The growth rate as modeled by Equation (4) was proposed in the literature [24] for the crystallization process that was utilized in this work.

$$G = k_{grw}^1 \left(\frac{c - c^*}{c^*}\right)^{k_{grw}^2} \tag{4}$$

where $k_{grw}^1$ is the growth kinetic coefficient, $k_{grw}^2$ is the order of kinetics, and $G$ is the growth rate (m/s). After transforming Equation (4) into Equation (5), the data points from growth experiments were plotted in Figure 8. The linear relation was correlated and the coefficients were found to be $k_{grw}^1 = 2.682 \times 10^{-3}$ and $k_{grw}^2 = 2.544$ in length direction, and $k_{grw}^1 = 6.072 \times 10^{-4}$ and $k_{grw}^2 = 2.520$ in width direction; the standard deviation was less than 0.103. The growth rate in thickness can be predicted by the measured mean ratio $c_t$ between thickness and width.

$$\ln(G) = \ln\left(k_{grw}^1\right) + k_{grw}^2 \cdot \ln\left(\frac{c - c^*}{c^*}\right) \tag{5}$$

In Goede's work [10], the growth rate was obtained via computing the average value of the growth rates in width and length directions, and assuming that the growth rate was a function of temperature. As a result, a relationship between the growth rate and temperature difference was built, as presented in Equation (6).

$$G(\text{m/s}) = 1.14 \times 10^{-5} \Delta T^{1.78} \tag{6}$$

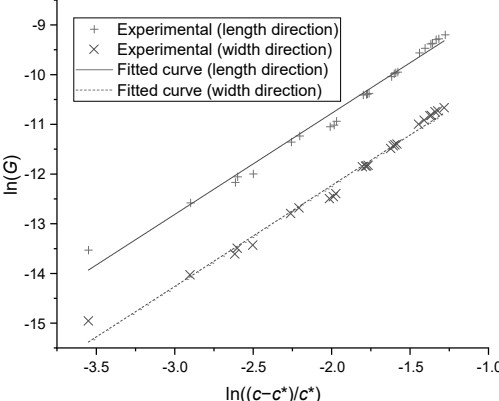

**Figure 8.** The growth rates versus supersaturations in particle length and width direction.

### 3.3. Breakage Kinetics

The broken crystals have irregular boundaries; therefore, the number change of broken crystals can be detected directly via OpenCV. The breakage kinetic rate was measured directly from the batch experiments, and studies on the topic usually indicate that breakage rate has a linear relation with particle weight [20]: $\frac{dm_j}{dt} = Km_j$, where $m_j$ is the weight of particles in $j$th discrete size range, $K$ is a constant, and $t$ is time. In this work, the thickness of a particle has fixed ratios with width. As a result, the weight of a particle can be represented: $m_j = c_t\rho w^2 l$, where $\rho$ is solid density, $w$ is width, and $l$ is length. The number of broken particles in a given size range can be represented by the breakage frequency $F^b$. Based on the above values, a new model was created as shown in Equation (7).

$$F^b = k_{brk}c_t w^2 l \tag{7}$$

where $k_{brk}$ is the constant breakage coefficient. The experimental data are plotted in Figure 9, in which the coefficient $k_{brk}$ regressed to be $7.086 \times 10^4$ with a relative standard deviation of 0.070. The particle was assumed to be broken into two parts with random size.

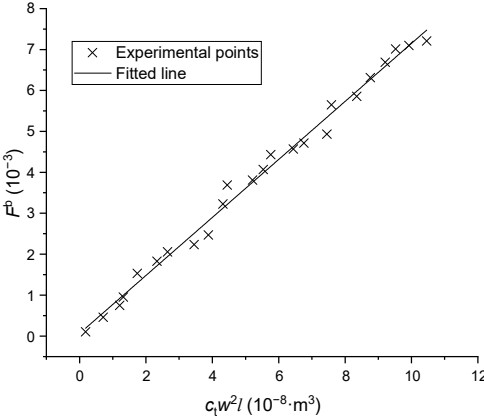

**Figure 9.** The breakage frequency versus particle volume.

### 3.4. Aggregation Kinetics

Based on Brownian motion, Smoluchowski [19] described the aggregation frequency between two particles, based on their distinct sizes and assuming all aggregation behaviors are binary in nature. The aggregation is also caused by particle growing, and the growth rate is a function of supersaturation. Therefore, Equation (8) was proposed in this work. $\sqrt[3]{c_t w^2 l}$ is the equivalent cube size, $k_{agr}$ is the coefficient and $F^a$ is the aggregation frequency. OpenCV can be used to detect the number of aggregated particles, which were distinct from normal and broken crystals; the data points from the aggregation experiments are

given in Figure 10. A linear relationship is present in the data, and the coefficient $k_{agr}$ was found to be $1.680 \times 10^{-5}$ with a relative standard deviation of 0.082.

$$F^a = k_{agr} \left( \frac{c - c^*}{c^*} \right) \left( \sqrt[3]{c_t w_1^2 l_1} + \sqrt[3]{c_t w_2^2 l_2} \right)^3 \tag{8}$$

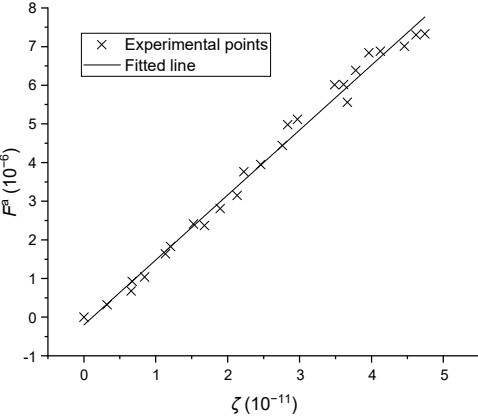

**Figure 10.** The aggregation frequencies versus operating conditions, where $\zeta$ represents $\left( \frac{c - c^*}{c^*} \right) \left( \sqrt[3]{c_t w_1^2 l_1} + \sqrt[3]{c_t w_2^2 l_2} \right)^3$.

## 4. Process Optimization

### 4.1. Process Equations and Analytic Solutions

The population balance equation (PBE) is the most commonly used model for the simulation of the continuous crystallization separation process. If a PBE is applied to describe the particle size distribution in a stirred crystallizer, the model can be expressed as [25,26]:

$$\frac{\partial n}{\partial t} + \frac{\partial (Gn)}{\partial L} + n \frac{\partial (\log V)}{dt} = B - D - \sum_i n_i S_i \tag{9}$$

where $n$ is the particle population distribution, which is a function of time and particle size; $t$ is operating time; $L$ is the particle size; the thickness of PX particle was assumed to have fixed ratio with width, hence, the PX particle can be represented only by width and length; $V$ is the volume of crystallizer, which is fixed in this work; $G$ is the growth rate of the crystal, which was just a function of supersaturation in this work; $B$ is the particle birth rate, which is derived from nucleation, breakage, and aggregation; $D$ is the particle death rate, which is caused by breakage; $S_i$ is the flow rate of inlet or outlet stream; and $n_i$ is the size distribution within the corresponding stream. As a result, Equation (9) can be simplified as:

$$\frac{\partial n}{\partial t} + G(w) \frac{\partial n}{\partial w} + G(l) \frac{\partial n}{\partial l} = B - D - \sum_i n_i S_i \tag{10}$$

The PBE is a partial differential equation, in which the size distribution $n$ is a function of time $t$ and particle size $w$ and $l$. The PBE is strongly non-linear, and as such does not possess an analytical solution in most cases. Therefore, a numerical solution should be used to solve it. There are three main numerical solutions proposed in the relevant literature [27]: the discretization method, the method of moment, and the finite element method. The discretization method, which involves moment of classes, is one of the most popular approaches for solving population balance equations [24]. Assuming the upper limits of the particle's width and length are $w^{max}$ and $l^{max}$, respectively, the size range can be discretized by intervals $\Delta w$ and $\Delta l$ into a matrix (see Figure 11). For an arbitrary cell $C_{w,l}$, $w$, and $l$ represent the width and length of the $n_{w,l}$ particles within it. At a given time $t$ and time interval $\Delta t$, new nucleated particles appear in $C_{1,1}$ at a rate of $B^0 \times \Delta t$. Then, the growth of particles in $C_{w,l}$ brings them into the green cells, for which the location can be calculated by

$(w + G(w) \times \Delta t)/\Delta w$ and $(l + G(l) \times \Delta t)/\Delta l$. The breakage of particles in $C_{w,l}$ generates particles with random size, which enter into the red cells, and the aggregation of particles therein form larger particles, some of which may enter $C_{w,l}$.

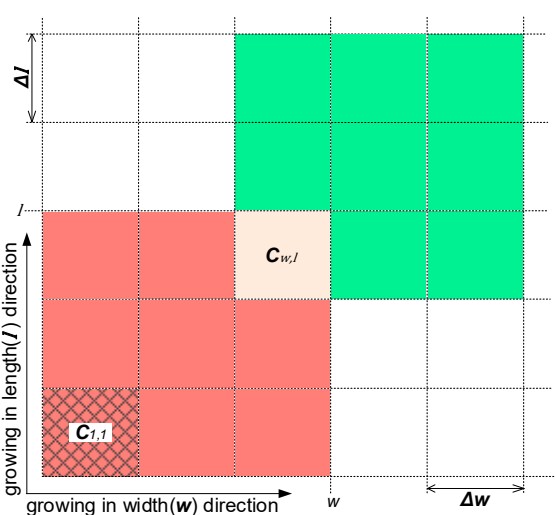

**Figure 11.** The discretization of particle width and length at time $t$.

The net flow of crystals in cell $C_{w,l}$ caused by growth is given in Equation (11) as follows:

$$
\begin{aligned}
n_{w,l}^{net} &= \Delta t \int_{w-\Delta w}^{w} \int_{l-\Delta l}^{l} \left( \frac{\partial[G(w)n_{w,l}]}{\partial w} + \frac{\partial[G(l)n_{w,l}]}{\partial l} \right) dl\, dw \\
&= \Delta t \frac{G(w)[n_{w,l}+n_{w+1,l}]-G(w-1)[n_{w-1,l}+n_{w,l}]}{2\,\Delta w} + \frac{G(l)[n_{w,l}+n_{w,l+1}]-G(l-1)[n_{w,l-1}+n_{w,l}]}{2\,\Delta l}
\end{aligned}
\tag{11}
$$

where $\frac{n_{w,l}+n_{w+1,l}}{2\,\Delta w}$ represents the average population density of particles in $C_{w,l}$ and $C_{w+1,l}$, and the equivalent applies to the other terms. The crystal flow fluxes at boundaries $w = 0$, $w = w^{max}$, $l = 0$, and $l = l^{max}$ are zero. The nucleation phenomenon is the appearance of new particles, which contributes to the increase in the quantity of particles in $C_{1,1}$, into which there are no inflows from the left and bottom cells. From the above, the net flow into $C_{1,1}$ can be written as below (Equation (12)):

$$
n_{1,1}^{net} = \Delta t \frac{G(w)[n_{1,1}+n_{2,1}]}{2\,\Delta w} + \Delta t \frac{G(l)[n_{1,1}+n_{1,2}]}{2\,\Delta l} + B^0 \Delta t
\tag{12}
$$

Particle breakage is a random event and, similarly, the resultant two particles are each of random size. For each particle in every cell, the breakage possibility $F^b$ was calculated using Equation (7), and a random value $\delta$ within [0, 1] was software generated. If $\delta$ was less than $F^b$, the particle split into two parts with random size and moved to corresponding cells. Aggregation is also a random event that occurs when two particles contact each other. For any pair, the possibility of aggregation $F^a$ was calculated by Equation (8), which was supplemented by a randomized value $\mu$ within [0, 1]. If $\mu \leq F^a$, two particles aggregated into a larger one and moved to the corresponding cell. The residence time was the average growth duration of particles in the crystallizer, which can be calculated by dividing the volume of the crystallizer by the flow rate.

The aforementioned algorithms were implemented in Matlab. To validate the reliability of the presented models, practical continuous crystallization experiments were carried out. The feed concentration and operating conditions for these experiments are given in Table 3, and the final particle volume distribution was determined by counting the number of particles within a given interval. The processes were also simulated by the developed subroutine while subjected to the same conditions. The particle width and length ranges were 2 mm and 8 mm, respectively; the size step was 0.02 mm, and the time step was 1 s.

The calculations continued until the size distribution in the crystallizer was stable, after which results indicated that the simulation time range of 30 min was enough. The program was executed on a Dell PC computer with 8 G memory and I5 CPU, so only a few minutes were required to finish the computation. The comparison of particle volume distributions between experimental results and calculated predictions is given in Figure 12, wherein the predictive data closely matched with the experimental results, thereby confirming the correctness of the proposed models.

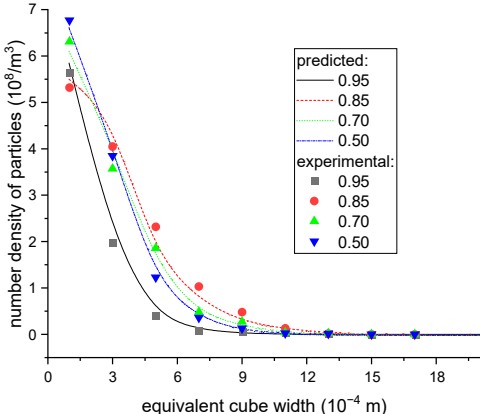

**Figure 12.** The equivalent cube size distributions of particles in the final products under different operating conditions.

### 4.2. Optimization of the Separation Process

The simplified flowsheet of isomerization reactor and multi-stage crystallizers (as presented in Figure 2) was a process with recycles and can be solved by equation-oriented or sequential-modular algorithm. For the three-stage crystallizers, the control variables were the crystallization temperature and the mean residence time in each crystallizer. However, the isomerization reactor, separator, and washing units were very complicated and not the subject of investigation in this work, so instead simplified models were used to simulate them. The target variables were the final yield of PX product and the energy needed in the process. The industrial product PX purity was typically 99.8%, and hence the crystallized solid particles were assumed to be pure PX. The reaction mechanism in the isomerization reactor was very complicated, and thus the composition of reaction and separation product (stream 4) were assumed to remain constant.

The material and composition balance equations for each stream are given below (Equation (13)), wherein $f_j$ is the flow rate of stream $j$; $x_j^i$ represents the composition of component $i$ in stream $j$; $n_j$ is the particle size distribution in stream $j$; and $F_r$, $F_s$, $F_c$, and $F_w$ are the calculation models for the isomerization reactor, separator, crystallizer, and centrifuge and wash units, respectively:

$$\begin{cases} f_1 x_1^i + f_{13} x_{13}^i = f_2 x_2^i = f_3 x_3^i = f_4 x_4^i + f_5 x_5^i \\ f_4 x_4^i + f_{14} x_{14}^i = f_6 x_6^i = f_7 x_7^i = f_8 x_8^i = f_9 x_9^i \\ f_9 x_9^i + f_{11} x_{11}^i = f_{10} x_{10}^i + f_{13} x_{13}^i + f_{14} x_{14}^i \\ f_{10} x_{10}^i = f_{11} x_{11}^i + f_{12} x_{12}^i \; and \; x_{10}^i = x_{11}^i = x_{12}^i \\ x_3 = F_r(x_2) \\ x_4 = F_s(x_3) \\ n_7 = F_c(x_6, T_1, t_r^1) \\ n_8 = F_c(x_7, T_2, t_r^2) \\ n_9 = F_c(x_8, T_3, t_r^3) \\ x_{14} = x_9^{liquid} \\ x_{13} = F_w(f_9, n_9, f_{11}) \end{cases} \quad (13)$$

For the separator model $F_s$, the PX, OX, MX, and ethyl benzene compounds were assumed to be separated from the mixture and sent to subsequent crystallizer units. The

presented models, as per previous sections, were used for solving $F_c$ in three crystallizers. The slurry with PX particles was introduced into the centrifuge unit, whereupon the PX-lean liquor and tiny-size PX particles (assumed to be of size < 0.2 mm) were separated from the resultant cake and recycled into the isomerization reactor. The remaining PX cake content was washed with the high-purity liquid PX product and centrifugally separated again. The PX-rich liquor was sent back into the first crystallizer, and the high-purity cake was introduced into the melting unit from which the final product was obtained. Referring to a practical Lummus industrial process, the impurity content in the wet PX cake (after centrifuging but before washing) was determined to be about 5 wt.%, the mass ratio between washing liquid and wet PX cake was 0.23, and only impurity contained in the wet PX cake was removed by washing. The PX cake was assumed to be pure after washing and centrifuging.

The process was simulated using a sequential-modular approach implemented in Matlab. Because of loops in the process, streams 10 and 14 were selected as the tear streams. Due to the complicated nature of the isomerization reactor and separator units, and the concentration of stream 4 being a function of the operating duration, temperature, and catalyst performance in the reactor, this aspect was outside of the scope of our investigation. Therefore, the isomerization reactor and separator were not accounted for in our simulation, and the data tabulated in Table 4 were used as the constant composition for stream 4. The compositions of streams 10 and 14 were updated after one round of calculations, which were continued if the difference between the previous composition and the new one was not within tolerance (relative deviation > $10^{-6}$); otherwise, the process converged. The model solving sequence is given in Figure 13, below:

**Table 4.** The composition of feed introduced to crystallizers in a practical industrial process.

| Compound | Fraction/wt.% |
|---|---|
| PX | 65.0 |
| OX | 6.0 |
| MX | 19.2 |
| Ethyl benzene | 5.3 |
| Others | 4.5 |

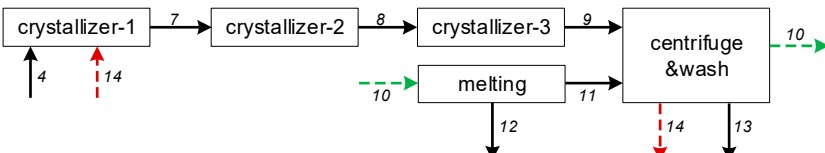

**Figure 13.** The solving sequence of models and the tear streams (10 and 14), where the numbers are the streams in Figure 2.

The ternary eutectic temperature was 207 K, as determined in previous investigations, which meant that the crystallization should be performed above this temperature. The operating temperature for each crystallizer (1, 2, and 3) were set at 265 K, 255 K, and 246 K, respectively, with a mean residence time of 10 min for each. Based on the variation of one of the six independent variables (temperatures and mean residence times across the three crystallizers), and keeping others fixed, the impact of operating conditions on product yield was given in Figures 14 and 15. The results indicate that the final yield of PX increases with residence time in the first and second crystallizers, and retains stability after 20 s. Conversely, while the yield increases with residence time in the third crystallizer, it remains stable after 300 s. Additionally, the product yield is not significantly affected by temperature, as in the first and second crystallizers, but increases rapidly with the reduction of temperature, as in the third.

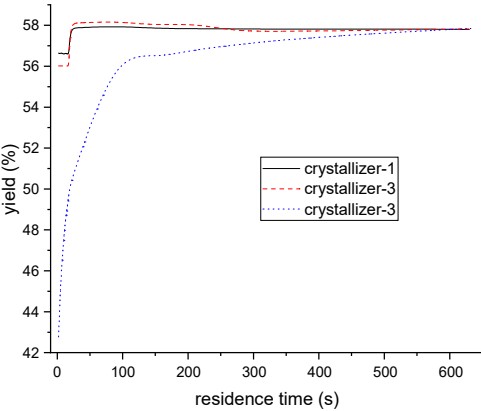

**Figure 14.** The relationship between PX yield and residence time of each crystallizer.

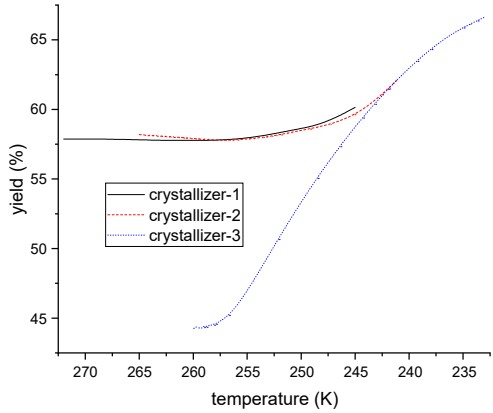

**Figure 15.** The relationship between PX yield and operating temperature of each crystallizer.

Although the above analysis was based on the simplified crystallization separation process and related parameters in this work, the proposed models and algorithms were still applied in other cases. For other continuous suspension crystallization separation processes with different compounds, operating temperatures, mean residence times, or stirring rates, the proposed models' parameters can be reproduced and no adjustment to the models is necessary.

## 5. Conclusions

The behaviors of nucleation, growth, breakage, and aggregation of PX particles were observed in an apparatus with both constant temperature and concentration, and the crystallization process was recorded via a microscope camera. The captured images were then transferred to a computer, where the boundaries of the particles were detected. The PX particle can be assumed to be a rectangular plate with fixed ratio between width and thickness. As a result, the width and length of the plate can be used to represent the three-dimensional size of a PX particle. The nucleation kinetic was built from the change in the number of particles at different temperatures and concentrations. The growth kinetic was correlated based on the plate size change in relation to time at different supersaturations. Finally, the breakage and aggregation models were regressed based on the observed particle collision and splitting.

A population balance equation was built to describe the mass balance and size distribution in the suspension crystallization separation crystallizers. Though only nucleation and growth kinetics were considered in most published papers as potential factors for solving the population balance equation, we included the breakage and aggregation kinetics in our work as well. Our population balance equation was solved using an extended moment of classes algorithm, and the solving process was implemented in MATLAB. Based on this, the number of particles in each moment class were updated using rigorous models.

From here, we designed and conducted a three-stage suspension crystallization separation experiment, in which each crystallizer had a distinct operating temperature and mean residence time. We also simulated the process using the aforementioned models, and the predicted results matched well with the experimental results, thus verifying the reliability of the proposed models. To enhance the yield of a multi-stage suspension crystallization separation process with loops, the effects of the operating temperature and mean residence time of each crystallizer on the final product yield were simulated. Ultimately, the distribution of product yield with crystallizer temperature and residence time was obtained, and the operating cost can also be calculated based on the crystallization duration and temperature. Although the simulation was carried out on a custom process, the proposed models and algorithms can still be applied in other cases and provide an alternative approach for optimizing continuous crystallization processes.

**Author Contributions:** Conceptualization, Z.C.; methodology, J.L.; validation, H.Z. and X.C.; formal analysis, H.Z.; investigation, Z.C.; resources, J.L.; data curation, J.L.; writing—original draft preparation, H.Z.; writing—review and editing, C.Y.; visualization, H.Z.; supervision, J.L.; project administration, C.Y.; funding acquisition, C.Y. All authors have read and agreed to the published version of the manuscript.

**Funding:** This research was funded by the National Natural Science Foundation of China, grant number 22078364.

**Data Availability Statement:** The data presented in this study are available on request from the corresponding author. The data are not publicly available due to partial experimental results being private.

**Conflicts of Interest:** The authors declare no conflict of interest.

## Nomenclature

| | |
|---|---|
| $B^0$ | nucleation rate, $\mathrm{m^{-3}s^{-1}}$ |
| $c$ | solution concentration |
| $c^*$ | saturated concentration |
| $c_\mathrm{t}$ | the ratios between width and thickness of PX particle |
| $F^a$ | aggregation frequency |
| $F^b$ | breakage frequency |
| $G$ | growth rate, m/s |
| $k_{agr}$ | coefficient for $F^a$ |
| $k_{brk}$ | constant breakage coefficient |
| $k_\mathrm{grw}^1,\ k_\mathrm{grw}^2$ | constants for $G$ |
| $k_\mathrm{nuc}^1,\ k_\mathrm{nuc}^2$ | constants for $B^0$ |
| $\Delta H_\mathrm{b}$ | enthalpy of vaporization, kJ/mol |
| $\Delta H_\mathrm{m}$ | enthalpy of melting, kJ/mol |
| $n$ | particle population distribution, $\mathrm{m^{-3}}$ |
| $t_r$ | the mean residence time, s |
| $T_\mathrm{b}$ | normal boiling point, K |
| $w,l$ | particle width and length, m |
| $T$ | temperature, K |
| $T_\mathrm{m}$ | melting point, K |

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
