# Peer review of "Modeling and Optimization of the Para-Xylene Continuous Suspension Crystallization Separation Process via a Morphology Technique and a Multi-Dimensional Population Balance Equation"

_processes, doi:10.3390/pr11030770_

Round 1

Reviewer 1 Report

All is clear!

Author Response

Review:

All is clear!

Reviewer 2 Report

The paper reviewed presents an interesting approach to follow the crystallization of p-xylene. The introduction is well written and the references are appropriate. However, I expected to see a simple comparison between the obtained values for the kinetic coefficients and those obtained in previous studies, a table of comparison will be very helpful for the readers, this comment joins your third point (What does it add to the subject area compared with other published material?).

This paper describes an interesting approach to follow the p-xylene crystallization kinetics by exploitation of images acquired with a microscope camera. The data acquired are plotted and they fit well with classical mathematical formalism e.g. Arrhenius equation. The nucleation, growth and breakage are considered to build a  population balance equation. Although the text is well written, the number of references may be increased to include previous studies using this approach (is there any?) and to compare the obtained results with previous ones obtained in different ways. The comparisons are missing. Also, the presentation of the setup using real photos either in the main text or in supporting information would be highly beneficial for the readers. Once these comments addressed, I propose to accept this paper for publication in  'Processes'.

Author Response

Review's point 1:

"I expected to see a simple comparison between the obtained values for the kinetic coefficients and those obtained in previous studies, a table of comparison will be very helpful for the readers", and "...compare the obtained results with previous ones obtained in different ways. The comparisons are missing..."

Response:

The kinetics and expression of the growth rate built by Geode were added into the last paragraph of Section 3.2.

The growth rate of para-xylene was only reported in Goede's work, however, he built a relationship between the average growth rate with the temperature difference, rather than the growth rates in each dimension with the supersaturation as given in our research work.

Review's point 2:

"....microscope camera.... the number of references may be increased to include previous studies using this approach"

Response:

The previous works adopted morphology approach and population balance equation were presented further in the penultimate paragraph of Introduction.

Review's point 3:

"the presentation of the setup using real photos either in the main text or in supporting information would be highly beneficial for the readers"

Response:

The photo of experimental equipment was added into Figure 3.

Reviewer 3 Report

Title: "Modeling and optimization of the para-xylene continuous suspension crystallization separation process via a morphology technique and a multi-dimensional population balance equation"

Reviewer comments/suggestions:

·  In the title the authors mention “…optimization…”

· Page 3, in “Introduction” it is written “…To address the gaps in existing literature, a new morphological approach was adopted to investigate the nucleation, growth, aggregation, and breakage behaviors…”

Reviewer suggestion: to highlight and enhance, in the introduction and conclusion, and to refer more precisely, the advantages and optimizations introduced by this work.

What are the optimizations introduced and the advantages - higher purity? better control and quality? more sustainable process? Lower temperatures? Less time consuming in the production?- of the methods developed in this work in comparison with the known and published methods.

Which “…gaps…” (introduction Pag 3) this works proposes to address, precisely?

Author Response

The research objects were presented in detail in the Introduction and Conclusion sections.

The advantages and optimization targets were discussed further in the Conclusion section.

The ambiguous statements in the Introduction section were removed.